Genome-wide evolution and expression analysis of the MYB-CC gene family in Brassica spp.

Gu Bin-Jie 1
Tong Yi-Kai 1
Wang You-Yi 1
Zhang Mei-Li 1
Ma Guang-Jing 2
Wu Xiao-Qin 1
Zhang Jian-Feng 1
Xu Fan 1
Li Jun 3
Ren Feng 1 renfeng@mail.ccnu.edu.cn
1 Hubei Key Laboratory of Genetic Regulation and Integrative Biology, School of Life Sciences, Central China Normal University , Wuhan, Hubei , China
2 Key Laboratory of Plant Germplasm Enhancement and Specialty Agriculture and Wuhan Botanical Garden, Chinese Academy of Sciences , Wuhan, Hubei , China
3 Key Laboratory of Biology and Genetic Improvement of Oil Crops, Ministry of Agriculture, Oil Crops Research Institute of the Chinese Academy of Agricultural Sciences , Wuhan, Hubei , China
Ansari Mahmood-ur-Rahman
Electronic publication date: 2022 Feb 25
Publication date: 2022
Volume: 10
Electronic Location ID: e12882
Received 2021 Jun 27; Accepted 2022 Jan 13
Copyright: © 2022 Gu et al.
Copyright year: 2022
Copyright holder: Gu et al.
License: This is an open access article distributed under the terms of the Creative Commons Attribution License, which permits unrestricted use, distribution, reproduction and adaptation in any medium and for any purpose provided that it is properly attributed. For attribution, the original author(s), title, publication source (PeerJ) and either DOI or URL of the article must be cited.
License URL: https://creativecommons.org/licenses/by/4.0/

Keywords: MYB-CC, Evolution, Brassica, Diploid, Allotetraploid

Funding: National Natural Sciences Foundation of China 31971814, 31571572 Project of National Research and Development of China 2016YFD0100202 The National Natural Sciences Foundation of China (Grant No. 31971814 and 31571572) and the Project of National Research and Development of China (Grant No. 2016YFD0100202) funded this research. The funders had no role in study design, data collection and analysis, decision to publish, or preparation of the manuscript.

==============================
The MYB-CC family is a subtype within the MYB superfamily. This family contains an MYB domain and a predicted coiled-coil (CC) domain. Several MYB-CC transcription factors are involved in the plant’s adaptability to low phosphate (Pi) stress. We identified 30, 34, and 55 MYB-CC genes in Brassica rapa, Brassica oleracea, and Brassica napus, respectively. The MYB-CC genes were divided into nine groups based on phylogenetic analysis. The analysis of the chromosome distribution and gene structure revealed that most MYB-CC genes retained the same relative position on the chromosomes and had similar gene structures during allotetraploidy. Evolutionary analysis showed that the ancestral whole-genome triplication (WGT) and the recent allopolyploidy are critical for the expansion of the MYB-CC gene family. The expression patterns of MYB-CC genes were found to be diverse in different tissues of the three Brassica species. Furthermore, the gene expression analysis under low Pi stress revealed that MYB-CC genes may be related to low Pi stress responses. These results may increase our understanding of MYB-CC gene family diversification and provide the basis for further analysis of the specific functions of MYB-CC genes in Brassica species.

Introduction

Phosphorus (P) is an essential macronutrient for plant growth and development. It plays a vital role in a number of processes, including metabolic regulation, energy transfer, and protein activation (Marschner & Marschner, 2012). P is transported into the plant from the soil, which is key to maintaining P levels in plant cells (Raghothama & Karthikeyan, 2005). P is abundant in many soils, however, plants utilize a phosphate (Pi) form that is very scarce (Holford, 1997). Consequently, plants often encounter Pi deficiency in agricultural and natural systems’ soils (Raghothama, 1999). In order to improve the absorption and usage of Pi in the soils, plants have evolved many adaptive responses to low Pi stress, known as Pi starvation response (PSR) (Raghothama, 1999; Williamson et al., 2001; Vance, Uhde-Stone & Allan, 2003). Morphologically, the architecture of the root system is altered under low Pi stress, which leads to a high root-shoot ratio and the root hair proliferation to enhance the total surface area for soil exploration (Williamson et al., 2001; Lopez-Bucio et al., 2002). Plants increase the availability of endogenous and exogenous inorganic Pi by increasing the P-replacing enzyme activity in metabolites and structural compounds and by releasing organic acids into soil solution (Duff, Sarath & Plaxton, 1994; Raghothama, 1999; Vance, Uhde-Stone & Allan, 2003). Furthermore, the root tips of some plants combine with mycorrhizal fungi to form mutually beneficial symbionts to modify soil scavenging potential (Harrison, 1999; Watt & Evans, 1999).

In Arabidopsis’ PSR, a subset of genes is induced or suppressed (Wu et al., 2003; Misson et al., 2005). Pi starvation-induced (PSI) genes may be regulated by several transcription factors, of which PHOSPHATE STARVATION RESPONSE 1 (PHR1) is the key transcription factor (TF) in vascular plants (Rubio et al., 2001; Devaiah, Karthikeyan & Raghothama, 2007; Devaiah et al., 2009; Ren et al., 2012). In Arabidopsis, PHR1 with a MYB domain and a predicted coiled-coil (CC) domain belongs to a 15-member family of MYB-CC transcription factors (Rubio et al., 2001). PHR1 and its homologues are also members of GARP subfamily transcription factors as they contain a consensus sequence (SHLQ(K/M)(Y/F)) similar to the motif (SHAQK(Y/F)F) found in MYB-related proteins (Du et al., 2013; Safi et al., 2017). PHR1 may bind to the specific cis-element (PHR1-binding sequence, P1BS) with an imperfect palindromic sequence GNATATNC in the promoter of PSI genes (Rubio et al., 2001). The Arabidopsis phr1 mutant displays a decreased expression of a subset of PSI genes and modified PSR, including a reduced Pi content and the accumulation of anthocyanin (Rubio et al., 2001; Misson et al., 2005). However, the overexpression of PHR1 in transgenic Arabidopsis increased the expression of a subset of PSI genes, the Pi content, and anthocyanin accumulation (Nilsson, Muller & Nielsen, 2007). The binding of PHR1 to P1BS elements in PSI gene promoters may be modulated by the SPX domain-containing proteins, such as SPX1 and SPX2, which interact with PHR1 and its orthologs to inhibit their activity, thereby suppressing PSR (Puga et al., 2014; Wang et al., 2014; Wild et al., 2016). PHR1 and the other MYB-CC members (PHL1, PHL2, PHL3 and PHL4) are crucial components of the central regulatory system controlling Arabidopsis’ transcriptional responses to Pi starvation (Bustos et al., 2010; Sun et al., 2016; Wang et al., 2018). Several PHR1 ortholog genes have been identified and characterized in different species (Valdes-Lopez et al., 2008; Zhou et al., 2008; Ren et al., 2012; Wang et al., 2013; Wang et al., 2019). The identification of PHR1 homologs and the establishment of their roles in these species have shown that central regulators have highly conserved functions. Furthermore, it suggests that members of the MYB-CC family may play a transcriptional regulatory role in controlling Pi uptake and homeostasis in plants. Therefore, it is crucial to identify MYB-CC genes and explore their regulatory roles in plant Pi uptake and homeostasis.

Brassica napus is a global essential oil crop. It is primarily cultivated for its healthy edible oil that is extracted from the seeds and its utility as a renewable biofuel, which is receiving increased attention. Vegetable types of the species have also been bred for consumption by both human and animals. B. napus is an allopolyploid (AACC, 2n = 38) resulting from natural interspecific hybridization between its two parental species: Brassica rapa (AA, 2n = 20) and Brassica oleracea (CC, 2n = 18) (Beilstein, Al-Shehbaz & Kellogg, 2006; Chalhoub et al., 2014). The genome evolution assay indicates the whole-genome triplication event in the Brassica lineage occurred about 13–17 million years ago (Ma) after it diverged from Arabidopsis thaliana about 20 Ma (Johnston et al., 2005; Town et al., 2006; Yang et al., 2006; Lysak et al., 2007; Mun et al., 2009). B. oleracea and B. rapa are also important vegetable crop species with a wide genetic and morphological diversity. B. rapa and B. oleracea differentiated from their ancestor approximately 3.75 Ma (Inaba & Nishio, 2002). The genome was doubled to form the heterotetraploid B. napus about 10,000 years ago (Cheung et al., 2009). Homologous fragment sequences in B. napus, B. rapa, and B. oleracea genomes indicate that there is a near-perfect microcollinearity between them (Cheung et al., 2009; Wang et al., 2011).

Our previous study revealed that BnPHR1 may play a crucial regulatory role in the Pi starvation response in B. napus (Ren et al., 2012). However, there is still a lack of research on other MYB-CC genes in B. napus and its parental species, B. rapa and B. oleracea. We conducted an evolutionary-based genome-wide analysis of the MYB-CC gene family in Brassica species. The gene structure, chromosome distribution, conserved motifs, phylogenetic relationship, cis-elements, and expression profiles were assayed systematically. These results may illuminate the evolutionary history of the MYB-CC gene family and assist in the investigation of the functions of MYB-CC genes in the Brassica species.

Materials and Methods

Plant materials and growth conditions

B. napus (accession Zhongshuang11) was kindly provided by the Oilcrops Research Institute, Chinese Academy of Agricultural Sciences. The seeds were disinfected in 70% ethanol for 1 min, washed with ddH2O three to four times, placed on a Petri dish lined with two layers of filter paper, and then germinated at 24 °C in growth chamber. After 3 days, the germinated seedlings were transferred to MS medium containing 0.3% agar and 3% sucrose at pH 5.7. The seedlings grew at 24 °C under a long-day light cycle (16 h light/8 h dark) and a relative humidity of 65%. Then the two-week-old plants were cultured in 1 mM and 1 μM Pi liquid medium for 24 h, respectively. Rapeseed seedlings’ shoots and roots were collected and frozen in liquid nitrogen. The RNA-seq analysis was carried out by Biomarker (Beijing). A total of 16 RNA samples were sequenced using the Illumina HiSeq 2,000 platform (Illumina, USA). The transcript abundance (FPKM value) was calculated based on the length of the gene and the reads mapped gene. The heatmap was constructed using TBtools (Chen et al., 2020). The raw RNA-seq data were deposited in the NCBI SRA database (https://ncbi.nlm.nih.gov/sra/) under the accession number PRJNA739537 and the assembled transcriptome data were deposited in the NCBI GEO database (https://www.ncbi.nlm.nih.gov/geo/)under the accession number GSE192452.

Identification of MYB-CC gene family in B. napus, B. rapa, and B. oleracea

The genes and proteins sequences in B. napus, B. rapa, and B. oleracea were downloaded from two databases (Genoscope: https://www.genoscope.cns.fr/brassicanapus/ and BRAD: http://brassicadb.cn/). The MYB-CC members were selected from B. napus, B. rapa, and B. oleracea by considering the E value < e−10 when compared with the 15 known MYB-CC protein sequences from A. thaliana (TAIR: http://www.arabidopsis.org/). This was achieved using BLAST on a computer (Rubio et al., 2001). The candidate protein sequences were further filtered using PFAM (http://pfam.xfam.org/) (Finn et al., 2014) and InterProScan (http://www.ebi.ac.uk/interpro) (Mitchell et al., 2015) according to the MYB and CC domains. The molecular weight and isoelectric point of MYB-CC members were calculated with ExPASy (Gasteiger et al., 2003).

Multiple sequence alignment and phylogenetic analysis of MYB-CC family

We constructed multiple sequence alignments of BnaMYB-CC, BraMYB-CC, BolMYB-CC, and AtMYB-CC using Clustal X (Thompson et al., 1997). The unrooted phylogenetic trees were constructed using the Neighbor-Joining (NJ) method. This was done using MEGA 6.0 software with the JTT model and pairwise gap deletion option (Tamura et al., 2013). The bootstrap analysis was conducted with 1,000 iterations.

Chromosomal location and gene duplication

The position of the MYB-CC genes was investigated according to the B. napus v4.1.chromosomal database (http://www.genoscope.cns.fr) and the Brassica database (BRAD, http://brassicadb.org). MapInspect software was used to locate the MYB-CC genes. The gene duplication was defined as: a coverage region between two sequences of more than 70% with more than 85% similarity in the coverage region (Zhou et al., 2004; Yang et al., 2008). Additionally, the ka (non-synonymous mutation rate) and ks (synonymous mutation rate) values of the repeating gene pairs were calculated using DnaSp software.

Gene structure and conserved motif analysis

GSDS (http://gsds.gao-lab.org/) was used to determine the exon/intron distribution of BnaMYB-CCs through the comparison of CDS and genomic sequences (Guo et al., 2007). The conserved motifs of BnaMYB-CCs were identified using MEME (Bailey et al., 2006) with the flowing parameters: the maximum number of motif: 10; the optimum width: 6–250; the number of repetitions: any. InterProScan was used to annotate the motifs (Quevillon et al., 2005).

Promoter elements analysis

From BRAD, a sequence of 2,000 bp upstream of the start codon was obtained for each MYB-CC gene and submitted to the PlantCARE website to determine the distribution of cis-elements in MYB-CC promoters (http://bioinformatics.psb.ugent.be/webtools/plantcare/html/) (Lescot et al., 2002).

Gene expression patterns in different tissues

The expression patterns of MYB-CC genes in different B. rapa, B. oleracea, and B. napus tissues were analyzed with previously published transcriptomic data (Tong et al., 2013; Yu et al., 2014; Sun et al., 2017). RNA-seq data were collected from the GEO database using accession numbers GSE43245 and GSE42891.

Results

Identification of MYB-CC genes in B. napus, B. rapa, and B. oleracea

The B. napus, B. rapa, and B. oleracea genomes were searched using the protein sequences of 15 MYB-CCs in A. thaliana. A total of 55 MYB-CC genes were identified in B. napus and were equally distributed in the An and Cn subgenomes (27 genes in An and 28 genes in Cn, respectively) (Table 1). A nomenclature system was used to distinguish the MYB-CC genes. The identified MYB-CC genes in B. napus were designated as BnaMYB-CC01 to BnaMYB-CC55 based on their order on An01, Cn01, An02, Cn02, etc.. The majority of BnaMYB-CCs contained 148 to 526 amino acids with molecular weights ranging from 16.991 kDa to 59.691 kDa. The predicted isoelectric points (pI) varied from 4.99 to 10.55 (Table 1). A total of 30 MYB-CC genes were identified in B. rapa and were designated as BraMYB-CC01 to BraMYB-CC30 based on their order on Ar01-Ann (Table S1). Furthermore, 34 MYB-CC genes were identified in B. oleracea and were assigned as BolMYB-CC01 to BolMYB-CC34 based on their order on Co01-Cnn (Table S2).

Table 1 MYB-CC gene family in B. napus.

Gene name	Gene symbol	Length (aa)	MW (Da)	pI	chr.Location	
BnaMYB-CC01	BnaA01g37390D	384	43,238.60	6.96	chrA01_random:2673192..2675057	
BnaMYB-CC02	BnaA01g08300D	417	45,819.52	6.08	chrA01:3948585..3951254	
BnaMYB-CC03	BnaA01g23340D	292	31,712.71	6.24	chrA01:15684038..15686754	
BnaMYB-CC04	BnaA01g30110D	437	48,643.16	5.75	chrA01:20712451..20715282	
BnaMYB-CC05	BnaA01g30340D	234	26,916.35	6.90	chrA01:20844513..20846132	
BnaMYB-CC06	BnaC01g09850D	403	44,256.62	6.03	chrC01:5804612..5807008	
BnaMYB-CC07	BnaC01g30360D	292	31,703.72	6.24	chrC01:28760606..28763683	
BnaMYB-CC08	BnaC01g38060D	437	48,633.04	5.63	chrC01:37146750..37149877	
BnaMYB-CC09	BnaA02g36130D	148	16,991.72	9.67	chrA02_random:724386..725308	
BnaMYB-CC10	BnaA02g14410D	324	36,771.36	6.29	chrA02:8128787..8130586	
BnaMYB-CC11	BnaA02g30980D	401	44,544.40	4.99	chrA02:22387811..22390602	
BnaMYB-CC12	BnaC02g01850D	376	42,385.10	6.72	chrC02:789792..792229	
BnaMYB-CC13	BnaC02g19330D	334	37,761.68	7.13	chrC02:15635506..15637277	
BnaMYB-CC14	BnaC02g39290D	314	35,590.55	5.07	chrC02:42253955..42255665	
BnaMYB-CC15	BnaA03g55680D	372	42,063.01	7.20	chrA03_random:375685..378053	
BnaMYB-CC16	BnaA03g32590D	432	48,203.65	5.71	chrA03:15741533..15744207	
BnaMYB-CC17	BnaA03g37160D	294	32,195.22	6.20	chrA03:18367120..18369001	
BnaMYB-CC18	BnaC03g73670D	422	47,150.68	5.63	chrC03_random:1699065..1702064	
BnaMYB-CC19	BnaC03g43430D	294	32,237.26	6.41	chrC03:28447863..28449907	
BnaMYB-CC20	BnaA05g37380D	381	42,773.13	6.37	chrA05_random:2969117..2971012	
BnaMYB-CC21	BnaA05g33010D	352	39,166.69	5.61	chrA05:22501618..22504921	
BnaMYB-CC22	BnaC05g40460D	231	26,542.02	7.23	chrC05:38626692..38627882	
BnaMYB-CC23	BnaC05g47270D	397	44,239.56	5.60	chrC05:42338193..42340293	
BnaMYB-CC24	BnaC05g47520D	372	41,833.12	6.51	chrC05:42526968..42528869	
BnaMYB-CC25	BnaA06g00030D	364	40,365.02	5.92	chrA06:15215..16810	
BnaMYB-CC26	BnaC06g19840D	343	38,387.30	8.76	chrC06:21970139..21972032	
BnaMYB-CC27	BnaC06g25290D	352	40,134.09	5.65	chrC06:26862555..26864876	
BnaMYB-CC28	BnaC06g28050D	334	37,763.62	7.12	chrC06:29356276..29358118	
BnaMYB-CC29	BnaC06g39850D	355	39,560.42	8.40	chrC06:36783795..36785668	
BnaMYB-CC30	BnaA07g00280D	338	37,838.63	6.01	chrA07:257201..258856	
BnaMYB-CC31	BnaA07g05970D	212	24,849.54	10.55	chrA07:6257304..6259312	
BnaMYB-CC32	BnaA07g20350D	343	38,340.27	8.76	chrA07:15976488..15978194	
BnaMYB-CC33	BnaA07g24220D	321	36,838.51	6.00	chrA07:18088750..18090258	
BnaMYB-CC34	BnaA07g28130D	331	37,499.33	7.12	chrA07:20352170..20354016	
BnaMYB-CC35	BnaA07g34920D	355	39,531.42	8.40	chrA07:23644702..23646584	
BnaMYB-CC36	BnaC07g00540D	340	38,246.02	5.67	chrC07:701526..703146	
BnaMYB-CC37	BnaC07g07340D	526	59,691.65	7.12	chrC07:11638655..11641909	
BnaMYB-CC38	BnaC07g18520D	280	31,234.19	7.72	chrC07:25179396..25180928	
BnaMYB-CC39	BnaC07g41450D	422	46,905.98	6.42	chrC07:41325045..41327724	
BnaMYB-CC40	BnaA08g13620D	396	43,731.26	5.75	chrA08:11763924..11766763	
BnaMYB-CC41	BnaC08g08750D	282	31,016.14	7.09	chrC08:13171439..13172925	
BnaMYB-CC42	BnaC08g13160D	398	43,966.59	5.81	chrC08:18195906..18198659	
BnaMYB-CC43	BnaA09g03540D	409	45,786.78	6.16	chrA09:1797716..1800204	
BnaMYB-CC44	BnaA09g10270D	317	35,908.38	5.13	chrA09:5256551..5258734	
BnaMYB-CC45	BnaC09g02840D	411	45,904.72	5.73	chrC09:1647123..1648856	
BnaMYB-CC46	BnaC09g54190D	393	44,222.36	6.00	chrC09_random:4002389..4004564	
BnaMYB-CC47	BnaC09g10430D	326	36,745.23	5.43	chrC09:7051680..7053999	
BnaMYB-CC48	BnaC09g48760D	335	37,665.00	8.33	chrC09:47473893..47476388	
BnaMYB-CC49	BnaA10g16590D	392	44,300.48	5.91	chrA10:12579079..12581358	
BnaMYB-CC50	BnaA10g24150D	374	42,216.30	7.66	chrA10:15810902..15813379	
BnaMYB-CC51	BnaAnng01860D	375	42,283.01	6.69	chrAnn	
BnaMYB-CC52	BnaAnng05640D	280	31,236.12	8.54	chrAnn	
BnaMYB-CC53	BnaCnng44310D	372	42,127.01	7.20	chrCnn	
BnaMYB-CC54	BnaCnng52990D	381	43,430.00	8.56	chrCnn	
BnaMYB-CC55	BnaCnng55690D	353	39,424.12	6.04	chrCnn	

Evolutionary analysis of MYB-CC family

To explore the evolutionary relationship of the MYB-CC gene family, an unrooted phylogenetic tree was generated using the Neiboring-Joining (NJ) method based on the full-length of 134 MYB-CC protein sequences from A. thaliana, B. napus, B. rapa, and B. oleracea. As shown in Fig. 1, the MYB-CC proteins were classified into nine distinct groups based on the branch of the phylogenetic tree. Group E and Group F each had 20 members, which were the larger groups. Group I was the smallest group with only two MYB-CC members. Group A, B, C, D, G, and H contained 19, 14, 14, 16, 13, and 14 members, respectively. One of the Arabidopsis MYB-CC members (coded by At2g01060) was not found to have a homolog in B. napus. The MYB-CC proteins in Group E were homologous to AtPHR1 and AtPHL4, and the members of Group B were homologs of AtPHL2 and AtPHL3. AtPHL1 belonged to Group F (Fig. 1).

Figure 1 Phylogenetic tree of MYB-CC proteins of A. thaliana, B. napus, B. rapa, and B. oleracea.

The phylogenetic tree was generated using the Neighbor-Joining (NJ) method implemented in the MEGA 6.0 software with JTT model and pairwise gap deletion option. The bootstrap analysis was conducted with 1,000 iterations.

Chromosomal distribution of MYB-CC genes

In order to determine the chromosomal distribution of MYB-CC genes, we searched the Brassica genome database and mapped MYB-CC genes to the corresponding chromosomes. The results showed that 50 BnaMYB-CC genes were distributed across 17 chromosomes except for An04 and Cn04 (Fig. 2). There were 27 BraMYB-CC genes located on nine chromosomes except for Ar04, and 28 BolMYB-CC genes were distributed on eight chromosomes except for Co04 (Fig. 2). Five BnaMYB-CC genes (BnaMYB-CC51, BnaMYB-CC52, BnaMYB-CC53, BnaMYB-CC54, and BnaMYB-CC55), three BraMYB-CC genes (BraMYB-CC28, BraMYB-CC29, and BraMYB-CC30), and six BolMYB-CC genes (BolMYB-CC29, BolMYB-CC30, BolMYB-CC31, BolMYB-CC32, BolMYB-CC33, and BolMYB-CC34) were mapped onto unanchored scaffolds (http://www.genoscope.cns.fr and http://brassicadb.org). In the An subgenome of B. napus, chromosome An07 carried the most (seven) MYB-CC genes. Chromosome An06 and An08 only contained one MYB-CC gene and chromosome An04 did not have MYB-CC genes. Comparatively, BnaMYB-CC genes were more prevalent in the Cn subgenome. Except for chromosome Cn04, the other chromosomes in the Cn subgenome contained two to four MYB-CC genes (Table S3).

Figure 2 Distribution of MYB-CC genes in genomes of B. napus, B. rapa, and B. oleracea.

The relative position of seven MYB-CC genes changed on the chromosomes after allotetraploidy. By evolving from diploid to allotetraploid, two BraMYB-CC genes were rearranged in the An subgenome belonging to B. napus. Comparatively, the positions of five MYB-CC genes in the Co and Cn were changed, of which three genes were on chromosome C06 (Fig. 2; Table S4). MYB-CC genes were located on several chromosomes, namely Ar03-An03, Ar09-An09, Ar10-An10, and Co08-Cn08. This was a conservative process compared to the gene distribution of B. napus and its diploid ancestors (Fig. 2). The results indicated that the genetic variation took place during the evolution of the B. napus genome from its diploid progenitors. The An subgenome was more stable than the Cn subgenome, which may be due to the more abundant homologous exchanges or richer transposable elements in the C subgenome (Chalhoub et al., 2014).

Collinearity analysis and gene duplication

Collinearity analysis of the MYB-CC genes was performed on B. napus and its two parental species (B. rapa and B. oleracea). As shown in Fig. 3, the collinear MYB-CC genes were widely distributed in the two subgenomes (An and Cn) and two genomes (Ar and Co), indicating that they promote evolution in the MYB-CC gene family. We also revealed the extension mechanism of the MYB-CC gene family in B. napus by studying gene duplication events. The results showed that 35 BnaMYB-CC genes formed 36 gene pairs with a high homology between gene and protein sequences. Some genes were involved in the duplication events more than once (Table 2). Among these segmental duplication events, 34 events happened between diverse chromosomes, and two events occurred on the same chromosome (BnaMYB-CC26 and BnaMYB-CC29 on Cn06; BnaMYB-CC32 and BnaMYB-CC35 on An07) (Fig. 2). Tandem duplication genes were described as a cluster of duplicated genes within 200 kb (Zhu et al., 2020). However, the two pairs of duplicated genes on the same chromosome were not closely related (Fig. 2; Table 1), suggesting that they were not the product of tandem duplication events.

Figure 3 Collinearity analysis of MYB-CC genes of B. napus, B. rapa, and B. oleracea.

Table 2 ka and ks of duplicated gene pairs.

Gene pairs	ks	ka	ka/ks	
BnaMYB-CC01	BnaMYB-CC20	0.2596	0.0519	0.1999	
BnaMYB-CC01	BnaMYB-CC24	0.2514	0.0386	0.1535	
BnaMYB-CC02	BnaMYB-CC06	2.2654	2.7285	1.2044	
BnaMYB-CC02	BnaMYB-CC40	0.3314	0.0599	0.1807	
BnaMYB-CC02	BnaMYB-CC42	0.3707	0.0710	0.1915	
BnaMYB-CC03	BnaMYB-CC07	0.0785	0.0091	0.1159	
BnaMYB-CC03	BnaMYB-CC17	0.3526	0.0588	0.1668	
BnaMYB-CC03	BnaMYB-CC19	0.3526	0.0588	0.1668	
BnaMYB-CC04	BnaMYB-CC08	0.0986	0.0222	0.2252	
BnaMYB-CC06	BnaMYB-CC40	0.3713	0.0689	0.1856	
BnaMYB-CC06	BnaMYB-CC42	0.4011	0.0834	0.2079	
BnaMYB-CC07	BnaMYB-CC17	0.3059	0.0673	0.2200	
BnaMYB-CC07	BnaMYB-CC19	0.2916	0.0673	0.2308	
BnaMYB-CC09	BnaMYB-CC26	0.4324	0.1823	0.4216	
BnaMYB-CC09	BnaMYB-CC29	0.2448	0.1710	0.6985	
BnaMYB-CC09	BnaMYB-CC32	0.4324	0.1823	0.4216	
BnaMYB-CC09	BnaMYB-CC35	0.2867	0.1749	0.6100	
BnaMYB-CC10	BnaMYB-CC13	0.0126	0.0150	1.1905	
BnaMYB-CC13	BnaMYB-CC34	0.3190	0.0684	0.2144	
BnaMYB-CC16	BnaMYB-CC18	0.1115	0.0269	0.2413	
BnaMYB-CC17	BnaMYB-CC19	0.1061	0.0090	0.0848	
BnaMYB-CC20	BnaMYB-CC24	0.1083	0.0099	0.0914	
BnaMYB-CC21	BnaMYB-CC23	0.1280	0.0290	0.2266	
BnaMYB-CC26	BnaMYB-CC29	0.2781	0.0420	0.1510	
BnaMYB-CC26	BnaMYB-CC32	0.0307	0.0152	0.4951	
BnaMYB-CC26	BnaMYB-CC35	0.2962	0.0394	0.1330	
BnaMYB-CC27	BnaMYB-CC33	0.0809	0.0467	0.5773	
BnaMYB-CC28	BnaMYB-CC34	0.0187	0.0078	0.4171	
BnaMYB-CC29	BnaMYB-CC32	0.2973	0.0435	0.1463	
BnaMYB-CC29	BnaMYB-CC35	0.0472	0.0073	0.1547	
BnaMYB-CC30	BnaMYB-CC36	0.0779	0.0237	0.3042	
BnaMYB-CC32	BnaMYB-CC35	0.3158	0.0409	0.1295	
BnaMYB-CC40	BnaMYB-CC42	0.1115	0.0077	0.0691	
BnaMYB-CC43	BnaMYB-CC45	0.1013	0.0485	0.4788	
BnaMYB-CC44	BnaMYB-CC47	0.1372	0.0653	0.4759	
BnaMYB-CC46	BnaMYB-CC49	0.1669	0.0122	0.0731	

The ka (non-synonymous mutation rate) and ks (synonymous mutation rate) were calculated using the DnaSP program to better understand the effects of evolutionary constraints on BnaMYB-CC genes in B. napus. The ratios of ka and ks of two gene pairs (BnaMYB-CC02 and BnaMYB-CC06; BnaMYB-CC10 and BnaMYB-CC13) were higher than one, indicating that the evolutionary process was actively selected and these genes may function redundantly, resulting in fast evolution (Table 2). However, the remaining 34 gene pairs’ ka and ks rations were less than one, indicating that most BnaMYB-CC genes were conservative and selected in the evolutionary process, resulting in a slower evolutionary rate (Table 2).

Gene structure and conserved motifs

To better understand the diversity of MYB-CC members, the gene exon/intron structure and conserved protein motifs were investigated. All BnaMYB-CC genes were interrupted by several introns, and the exon/intron distribution was complicated (Fig. 4A). BnaMYB-CC37 contained the most introns (11), while several members in group A had the fewest introns (BnaMYB-CC05, BnaMYB-CC09, and BnaMYB-CC22 with four introns each). The members belonging to the same group showed similar exon/intron arrangements (Fig. 4A). For example, the BnaMYB-CC genes in Group H had five introns with similar positions and lengths, suggesting they may have resulted from gene duplication.

Figure 4 Phylogenetic analysis, exon/intron organization and conserved motifs of MYB-CC genes.

Forty-five pairs of homologous genes, which have the closest genetic distance, were analyzed to further investigate the variation of gene structure after allotetraplpidy. The results indicated that the genes were clustered together at the terminal level of the phylogenetic tree (Table S5). Seven pairs of homologous genes showed intron loss/gain variations, while the remaining 38 pairs had the same number of introns (Table S5). In contrast to their ancestral genes, most BnaMYB-CC genes had an identical exon/intron phase (Fig. 4A). It should be noted that intron acquisition events mainly occurred in group C. Specifically, BnaMYB-CC01, BnaMYB-CC20, and BnaMYB-CC54 had one more intron than their homologous genes in the diploid progenitors (Table S5). The majority of the homologous pairs retained consistent number and a similar phase of exon/intron, indicating that MYB-CC gene structures was conserved.

Ten conserved motifs (motifs 1–10) with lengths ranging from 15–50 amino acids were identified using the MEME program (Fig. 4B). The motifs were annotated using the InterProScan program. The results showed that the MYB-like DNA-binding domain and the predicted coiled-coil domain (motif 1 and 2) were found in all BnaMYB-CCs (Fig. 4C). Moreover, similar to the exon/intron organization, the members belonging to the same group also showed similar motif composition. Motif 6 and 7 were found uniquely in the BnaMYB-CC proteins in Group D. Only the proteins of Group A contained motif 8, while motif 10 was found mainly in Group C (Fig. 4C). The other six motifs were widely distributed among the groups. For example, motif 9 was absent in Group A and Group D (Fig. 4C) indicating that there were similarities in the conserved sequences of different phylogenetic groups.

Cis-elements in promoters of MYB-CC genes

To investigate the cis-elements in the promoters of MYB-CC genes, a 2 kb sequence upstream of the start codon of each gene was extracted from the B. napus genome. The promoter sequences were submitted to PlantCARE. As shown in Fig. 5 and Table S6, two types of cis-elements (phytohormone-related and abiotic stress-related elements) were selected. Methyl jasmonate (MeJA)-responsive element (TGACG) and ABA-responsive elements (ABRE) were present in 78.1% and 74.5% of promoters of BnaMYB-CCs, respectively. In the promoters of 55 BnaMYB-CCs, 21 contained auxin-responsive elements (TGA-element, AuxRE, and AuxRR-core), 26 contained SA responsive element (TCA element), and 27 contained gibberellin-responsive elements (GARE-motif, TATC-box, and P-box). Two cis-lements (TC-rich repeats and LTR, related to abiotic stress) were widely distributed in BnaMYB-CC promoters. BnaMYB-CC20 had seven LTRs in its promoter (Fig. 5), indicating that BnaMYB-CC20 may play a vital role in response to low-temperature stresses. The results suggest BnaMYB-CCs may be involved in stress resistance and the hormone signaling pathway. It is worth noting that the cis-elements related to light response were found in significant quantity in the promoters of BnaMYB-CC. However, our work focused on the cis-elements related to phytohormones and thus, light responsive cis-elements are not shown in Fig. 5.

Figure 5 Cis-elements analysis of BnaMYB-CC genes.

The 2 kb sequences upstream from the transcription start site were investigated. Different colored boxes represent different Cis-acting elements.

Expression profiles of MYB-CC genes in different tissues

To further understand the biological functions of MYB-CCs, the expression profiles of the genes in different tissues of B. napus, B. rapa, and B. oleracea were assayed based on the transcriptomic data. As shown in Fig. 6, MYB-CC genes were expressed widely in diverse tissues. There was a high accumulation of BraMYB-CCs transcripts found in the roots (Fig. 6A; Table S7). BraMYB-CCs in Group A (BraMYB-CC06, BraMYB-CC17, BraMYB-CC20, and BraMYB-CC28) and Group C (BraMYB-CC03, BraMYB-CC12, and BraMYB-CC26) were extensively expressed in tissues except for silique (Fig. 6A). The transcripts of BolMYB-CCs were generally found in callus (Fig. 6B; Table S7). In B. napus, BnaMYB-CC05, BnaMYB-CC06, BnaMYB-CC19, BnaMYB-CC40, and BnaMYB-CC42 were significantly expressed in all tissues (Fig. 6C). The majority of BnaMYB-CC genes in Group A (BnaMYB-CC09, BnaMYB-CC26, BnaMYB-CC29, BnaMYB-CC32, and BnaMYB-CC35) and Group C (BnaMYB-CC01, BnaMYB-CC20, BnaMYB-CC46, BnaMYB-CC49, and BnaMYB-CC54) showed a low level of expression in the ovule, while their transcripts were found in other tissues (Fig. 6C). BnaMYB-CC12, BnaMYB-CC15, BnaMYB-CC27, BnaMYB-CC33, BnaMYB-CC48, and BnaMYB-CC55 were expressed in fewer tissues (Fig. 6C).

Figure 6 Expression patterns of MYB-CC genes in tissues.

(A) Expression levels of 30 BraMYB-CC genes in six tissues of B. rapa. (B) Expression levels of 32 BolMYB-CC genes in seven tissues of B. oleracea. (C) Expression levels of 55 BnaMYB-CC genes in 12 tissues of B. napus.

Expression analysis of MYB-CC genes under low Pi stresses

The expression profiles of MYB-CCs under low Pi stress were analyzed by RNA-Seq data to identify potential MYB-CCs with regulatory roles in Pi starvation responses. The original FPKM values of the transcriptome are shown in Table S8. A heatmap was constructed to display diverse expression levels in the roots and shoots under low or high Pi conditions (Fig. 7). The expression of most MYB-CC genes was more active in the roots than that in the shoots, especially BraMYB-CCs and BolMYB-CCs (Fig. 7). The expression patterns of several genes were altered after allopolyploidy. For example, BraMYB-CC24, BolMYB-CC16, and BolMYB-CC31 were mainly expressed in the roots, while their orthologous genes BnaMYB-CC44, BnaMYB-CC26, and BnaMYB-CC23 were expressed dramatically in the shoots (Fig. 7). However, other genes retained the same expression pattern after allopolyploidy, implying that the BnaMYB-CC genes were conserved and maintained similar functions in diploid progenitors (Fig. 7; Table S8).

Figure 7 Expression patterns of MYB-CC genes in response to low Pi stresses.

(A) Expression levels of 30 MYB-CC genes in B. rapa. (B) Expression levels of 34 MYB-CC genes in B. oleracea. (C) Expression levels of 55 MYB-CC genes in B. napus. HP, high Pi condition; LP, low Pi condition. The heat map was created using TBtools.

Approximately half of the BnaMYB-CCs were induced by low Pi stress in both roots and shoots (Fig. 7). Notably, some BnaMYB-CCs were induced twofold or more in roots (BnaMYB-CC01, BnaMYB-CC32, and BnaMYB-CC44) or shoots (BnaMYB-CC04, BnaMYB-CC10, BnaMYB-CC16, and BnaMYB-CC18) under low Pi conditions (Table S8). In B. rapa and B. oleracea, approximately one-third of BraMYB-CCs and two-thirds of BolMYB-CCs genes were up-regulated under low Pi stress in roots (Fig. 7; Table S8). In B. napus, BnaMYB-CCs of Group D (BnaMYB-CC12, BnaMYB-CC15, BnaMYB-CC48, BnaMYB-CC50, BnaMYB-CC51, and BnaMYB-CC53) were significantly down-regulated in both roots and shoots under low Pi stress (Fig. 7C; Table S8). Seven BnaMYB-CCs of Group A were moderately induced by low Pi stress in roots, while their expression levels increased in shoots under high and low Pi conditions (Fig. 7; Table S8). However, BnaMYB-CC10, BnaMYB-CC13, BnaMYB-CC27, and BnaMYB-CC34 of Group H were induced by low Pi stress in roots prominently. All six members of BnaMYB-CCs in Group H were induced by low Pi stress in shoots (Fig. 7; Table S8). The results indicate that the expression patterns were conservative for MYB-CCs of the same evolutionary group.

Discussion

The MYB-CC family is a subtype within the MYB superfamily. It is contains the MYB domain and the predicted coiled-coil (CC) domain. The MYB-CC family has 15 members in A. thaliana (Rubio et al., 2001). PHR1, PHL1, PHL2, and PHL3 have been redundantly involved in regulating Pi starvation responses (Rubio et al., 2001; Bustos et al., 2010; Sun et al., 2016). PHL4 is one of the MYB-CC in Arabidopsis, and was regarded as an unimportant regulator in responses to Pi starvation (Wang et al., 2018). MYB-CCs have also been shown to be involved in plant growth regulation. One member of the MYB-CC Group A, APL, plays a dual role in inhibiting xylem differentiation and promoting phloem differentiation during vascular development (Bonke et al., 2003). MYR1 and MYR2 of MYB-CC Group C are redundant negative flowering time regulators under low light induction (Zhao et al., 2011). GFR of MYB-CC Group D is a low-temperature regulator of flavonoid accumulation (Petridis et al., 2016). The MYB-CC family was also mentioned in other plant species, such as 12 ZmMYB-CCs in maize (Bai et al., 2019) and 13 FvMYB-CCs in woodland strawberry (Wang et al., 2019). MYB-CC genes in wheat have been reported in response to drought stress (Li et al., 2019c). Additionally, BnPHR1, which is a vital regulator of low Pi responses, was isolated and identified in B. napus (Ren et al., 2012). The comprehensive identification and evolutionary analysis of the MYB-CC family in B. napus and its parental species B. rapa and B. oleracea have not been reported to date. We comprehensively analyzed the MYB-CC family in Brassica species and assayed their expression patterns under low Pi stress.

Gene duplication is the main driving force for gene family expansion (Taylor & Raes, 2004). Brassica species shared multiple paleo-polyploidy (whole-genome duplication) events with A. thaliana, providing primitive genetic material for biological evolution and facilitating the adaption to the changing environments (Bowers et al., 2003). Compared to the 15 MYB-CC genes in Arabidopsis, the number of MYB-CC genes in B. napus (55), B. rapa (30), and B. oleracea (34) increased significantly. This is due to the additional whole-genome triplication (WGT) event in the Brassica species after its differentiation from A. thaliana (Lysak et al., 2005; Town et al., 2006; Wang et al., 2011). B. napus is a heterotetraploid formed by hybridization between B. rapa and B. oleracea, followed by chromosome doubling (Cheung et al., 2009; Chalhoub et al., 2014; Wu et al., 2019). The segmental and tandem duplication events contribute to the gene family expansion during evolution (Cannon et al., 2004). We found that 35 BnaMYB-CC genes formed 36 pairs with high gene and protein sequence identity and similarity. All of these gene pairs were identified as segmental repeats, indicating that the amplification of the MYB-CC gene family in B. napus may be independent of tandem duplication. The same duplication pattern was reported in other gene families during the evolution from diploid to allotetraploid (Li et al., 2019a; Li et al., 2019b; Wang et al., 2020), and the gene families were mainly influenced by segmental duplication resulting from WGT and allopolyploidy (Li et al., 2017). Specifically, plants retain many duplicate chromosome blocks in their genomes due to polyploidization events, resulting in segmental duplication (Cannon et al., 2004). Tandem repeats are the result of unequal crossing-over between similar alleles (Achaz et al., 2000). Although no tandem duplication was found in this study, it is still an important mechanism for the expansion of gene family in plants. For instance, tandemly arrayed genes comprise more than 10% of the genes in A. thaliana genome (Rizzon, Ponger & Gaut, 2006).

Approximately 45 MYB-CC genes were predicted to exist in both B. rapa and B. oleracea as a result of the additional WGT in Brassica species. However, only 30 and 34 MYB-CC genes occurred in two diploid species and large-scale gene loss after WGT, respectively (Cheng, Wu & Wang, 2014). The B. rapa genome contains approximately twice the number of genes of A. thaliana due to genomic shrinkage and differential loss of duplicated genes after WGT events (Mun et al., 2009). It has been previously reported that 35% of genes in Brassica were lost by a deletion mechanism when WGT occurred (Town et al., 2006). The concentration of some gene products were changed after WGT, which may lead to an imbalance of gene dose. The relatively low retention frequency of these dose-changed genes may also explain the genetic loss after WGT (Freeling, 2008). The number of MYB-CC genes in B. napus was found to be less than the sum of its two diploid ancestors, indicating gene loss also happened after allopolyploidization. The 13 deleted genes were listed in Table S4, of which five genes belong to the A subgenome and eight belong to the C subgenome. This may be due to the rearrangement of genome sequences after hybridization (Paterson, Bowers & Chapman, 2004). In general, each of MYB-CCs in Arabidopsis was expected to have six homologs in B. napus. However, 12 MYB-CCs of Arabidopsis had less than six homologous genes in B. napus. For example, MYR1 had only two homologs (BnaMYB-CC46 and BnaMYB-CC49), which may be caused by the loss of unnecessary duplicates during evolution. The retained MYB-CC gene replications may play a non-redundant role in B. napus.

MYB-CC transcription factors are mainly involved in the responses to Pi starvation (Rubio et al., 2001; Nilsson, Muller & Nielsen, 2007; Zhou et al., 2008; Valdes-Lopez et al., 2008; Bustos et al., 2010; Ren et al., 2012; Wang et al., 2013; Sun et al., 2016; Ruan et al., 2017; Wang et al., 2018; Wang et al., 2019). Data on MYB-CC genes in B. napus are limited; however, the homologous AtMYB-CCs may predict the functions of BnaMYB-CCs in the same phylogenetic group. In Arabidopsis, PHL2 is modulated by Pi starvation and is redundant with PHR1 in regulating the responses to Pi starvation (Sun et al., 2016). The homologs of PHL2 in B. napus (BnaMYB-CC03, BnaMYB-CC07, BnaMYB-CC17, and BnaMYB-CC19) were only slightly induced in shoots under low Pi conditions (Fig.7; Table S8), suggesting that the functions of the genes might have changed. The transcription of AtPHR1 was not significantly regulated by Pi starvation (Rubio et al., 2001) and its homologous genes (BnaMYB-CC02, BnaMYB-CC06, BnaMYB-CC39, BnaMYB-CC40, and BnaMYB-CC42) in B. napus were not induced by low Pi stresses either (Fig.7; Table S8). The homologous genes of AtPHR1 were also inactive at the transcriptional level under low Pi stress in other plant species (Zhou et al., 2008; Wang et al., 2019). Our results support that post-translational modification mainly regulates the activity of PHR1 (Miura et al., 2005). To date, the functions of the majority of MYB-CC genes in Arabidopsis and in other plant species are still unclear. Under low Pi stress, MYB-CC genes are mainly involved in anthocyanin biosynthesis and accumulation, Pi redistribution and homeostasis in plant shoots, and involved in modification of root system architecture and Pi uptake in plant roots. The differentiation of the expression patterns of MYB-CC genes from diploid to allotetraploid suggests that their role assignments in above mentioned processes could be changed. However, the functions of each MYB-CC gene in plant Pi homeostasis and low Pi response need to be further studied.

To further characterize the MYB-CC genes in responses to Pi starvation evolutionarily, we selected 38 pairs of potential orthologous genes for expression analysis. The expression levels of BnaMYB-CCs were significantly lower than in diploid species (Table S9). Some gene pairs with the same location had different expression patterns, whereas others with different location shared similar expression patterns (Table S9). There may be no direct correlation between the relative position of the gene and its expression patterns.

Conclusions

This study comprehensively evaluated the identification, classification, expression and evolution analyses of MYB-CC gene family of Brassica species. A total of 30, 34, and 55 MYB-CC genes were identified in B. rapa, B. oleracea, and B. napus, respectively. All of the MYB-CC genes were divided into nine groups in the phylogenetic tree. Members of the same group have similar gene structures and conserved motifs, indicating that they could be conserved during evolution. In Brassica, WGT and allotetraploidy were vital for the expansion of MYB-CC genes. Gene loss occurred widely for a number of reasons during the evolutionary process. An analysis of gene expression under low Pi stress showed that there was no significant relationship between the relative positions of MYB-CC genes and their expression patterns. This work will promote the understanding of the MYB-CC gene family and assist further analysis of the specific functions of MYB-CC genes in the Brassica species.

Supplemental Information

Supplemental Information 1 MYB-CC gene family in B. rapa.

Click here for additional data file.

Supplemental Information 2 MYB-CC gene family in B. oleracea.

Click here for additional data file.

Supplemental Information 3 The number of MYB-CC genes in every chromosome in B. napus, B. rapa and B.oleracea.

Click here for additional data file.

Supplemental Information 4 The selected differential genes in B. napus, B. rapa and B. oleracea.

Click here for additional data file.

Supplemental Information 5 Cis-elements in the promoters of MYB-CC genes.

Click here for additional data file.

Supplemental Information 6 The FPKM values of BraMYB-CC genes in RNA-Seq analysis.

Click here for additional data file.

Supplemental Information 7 Intron information of MYB-CC gene pairs with the closest evolutionary relationships.

Click here for additional data file.

Supplemental Information 8 The FPKM values of BnaMYB-CC genes in RNA-Seq analysis.

Click here for additional data file.

Supplemental Information 9 The expression comparison of MYB-CC genes with relative position changed/maintained.

Click here for additional data file.

Supplemental Information 10 RNAseq dataset of B. napus.

Click here for additional data file.

Supplemental Information 11 RNAseq dataset of B. oleracea.

Click here for additional data file.

Supplemental Information 12 RNAseq dataset of B. rapa.

Click here for additional data file.

Additional Information and Declarations

Competing Interests

Author Contributions

DNA Deposition

Data Availability

The authors declare that they have no competing interests.

Bin-Jie Gu conceived and designed the experiments, performed the experiments, analyzed the data, prepared figures and/or tables, and approved the final draft.

Yi-Kai Tong performed the experiments, analyzed the data, prepared figures and/or tables, and approved the final draft.

You-Yi Wang performed the experiments, prepared figures and/or tables, authored or reviewed drafts of the paper, and approved the final draft.

Mei-Li Zhang conceived and designed the experiments, performed the experiments, analyzed the data, prepared figures and/or tables, and approved the final draft.

Guang-Jing Ma performed the experiments, authored or reviewed drafts of the paper, and approved the final draft.

Xiao-Qin Wu performed the experiments, authored or reviewed drafts of the paper, materials, and approved the final draft.

Jian-Feng Zhang performed the experiments, authored or reviewed drafts of the paper, and approved the final draft.

Fan Xu performed the experiments, authored or reviewed drafts of the paper, materials, and approved the final draft.

Jun Li analyzed the data, prepared figures and/or tables, and approved the final draft.

Feng Ren conceived and designed the experiments, analyzed the data, prepared figures and/or tables, and approved the final draft.

The following information was supplied regarding the deposition of DNA sequences:

The raw reads from this study are available at NCBI: PRJNA739537.

The following information was supplied regarding data availability:

The RNA-seq data are available in the Supplemental Files and NCBI: PRJNA739537.

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
