# Peer review of "Genome-wide evolution and expression analysis of the MYB-CC gene family in Brassica spp"

_PeerJ, doi:10.7717/peerj.12882_

## Round 0.1 · original submission · Minor Revisions

As per reviewers' suggestions, the paper needs "Minor Revision". Please address all the comments raised by reviewers, especially Reviewer-2. Provide a point-to-point response to the questions asked.

Reviewer 1 ·

Basic reporting

no comment

Experimental design

no comment

Validity of the findings

no comment

Additional comments

The manuscript from Gu et al. Comprehensively analyzed MYB-CC gene family in Brassica species. MYB-CC family members, including PHR1, play significant roles in plant responses to low Pi stresses and other biological processes. The data in this manuscript will contribute to understand the evolutionary history of the MYB-CC gene family and their biological functions in Brassica species. I do have some minor concerns that I believe should be addressed prior to publication.
1. In Fig. 5, two types of cis-elements (hormone-related and abiotic stress-related elements) should be provided the specific element sequences respectively in Results, such as the MeJA-responsive elements TGACG.
2. Line 290-291, “methyl jasmonate (MeJA)-responsive elements (TGACG-motif and CGTCA-motif)” should be revised as “methyl jasmonate (MeJA)-responsive elements (TGACG)”.
3. Line 325, “expression profiles of MYB-CC under Pi stresses” should be revised as “expression profiles of MYB-CC under low Pi stresses”.
4. In Fig. 7, the inducible MYB-CCs by low Pi should be analyzed in detailed among B. napus, B. rapa and B, oleracea.
5. Manuscript needs to be corrected by professional. There are some grammatical errors in current version.

·

Basic reporting

The manuscript “Genome-wide identification and evolutionary analysis of MYB-CC gene family in two Brassica diploid species and their allotetraploid relative” (Ms ID: #61732v1) by Gu et al. presents a genome-wide identification and expression analysis of the PHR1 homolog genes in in Brassica rapa, Brassica oleracea and Brassica napus genomes. It is a survey study. The authors identified and investigated this gene family in these three genomes and did a bunch of bioinformatic analyses of the candidates. They also explored the expressions of the candidates in different tissues and under low Pi stresses. The reason to perform this study is convictive. The methods are relevant and well applied for most of the study. This type of study falls within the scope of Peerj. Overall, the contents and quality of this ms is suitable to be published in this journal. But, some points listed below should be corrected before it could be accepted for publication.


Major revision:
1. The title of this ms is a little long. It is better to compress it, such as “Genome-wide evolution and expression analysis of MYBCC gene family in Brassica rapa, Brassica oleracea and Brassica napus”.

2. In my opinion, it is worthy to discuss/check the nomenclature of this type of genes. They may be classified into the GARP-like gene family, which are often confused with MYB-related genes, because they contain a consensus sequence (SHLQKY) very similar to that of CCA1-like proteins (SHAQK(Y/F)F) at the C-terminal of the BD domain. However, they contain only 1 of the 3 regularly spaced Trp (W) residues found in the MYB domain of the typical MYB proteins (Du et al., 2013, DNA research).

3. “Abstract”, the contents in this section are poorly organized and wrote. The authors should focus on the main results of this study in this section, and state the results accurately to avoid confusion. For example, line 22-27, the background of this study is too long. Similarly, it is better to summarize the main results in a shorter length in line 29-34. The content in line 30 is not your result, it is just the background. So, it is better to delete it. In addition, line 34-36, the expression profile of this gene family is not clear. And the results related to the expressions under low Pi stresses were not showed here?

4. Line 51-53, the statement is confusing. The architecture of root system will be altered under which environmental condition? Low Pi stresses? Similar problems should be checked in the full text carefully.

5. Line 98, the statement is confusing as it should be named as sub-genome in B. napus. In addition, “It” should be “it”.

6. Line 112, “Chinese academy of agricultural sciences” should be “Chinese Academy of Agricultural Sciences”?

7. Line 113-123, as mentioned in the following text, the expression of candidate genes under low Pi stresses were analyzed based on the public RNA-Seq datasets acquired from NCBI? So, please re-write this part.

8. Line 128-130, please show the names of these three databases.

9. Line 158, please supplied the website of the “GSDS” online software.

10. Line 170-171, the content in this sentence is repeated with that in line 111-112?

11. The section of “Evolutionary analysis of MYB-CC transcription factor family”, the statement in this part is confusing. As the NJ tree was constructed based on the protein sequences, you should use “protein” instead of “gene” in the text. Accordingly, it is better to use the protein name in the text. Moreover, please state the reason/ criterion to classify the MYB-CC genes into nine distinct groups.

12. Line 217, please state the detail number of the MYB-CC genes, instead of using “several”.

13. Line 230-231, please delete the sentence of “Two genes linked together by a red line were collinear genes”.

14. Line 245-248, in general, the ratio of ka and ks of duplicated genes is <1. It is interesting that there are two gene pairs (BnaMYB-CC02/BnaMYB-CC06; BnaMYB-CC10/BnaMYB-CC13) were > 1. Please check the sequences of these two gene pairs to avoid genomic quality issues firstly.

15. Line 256, “and” needs not to be in italic.

16. Line 300-301, the authors stated that “It is worth noting that most cis-elements in the promoters of BnaMYB-CC genes were mainly 301 related to light response, but they were not shown in Fig. 5”. Please state the reason why you did not show this result.

17. Line 348, “their expressing level is” should be “their expressing levels were”?

18. The contents in the section of “Conclusions” is a little long, it is better to compress it and summarize the main conclusion of this ms.

19. Table 1, please unify the decimal point digit in this table.

20. Figure 1, please supply the bootstrap value on the NJ tree for each group. And the names of the four species should be in italic in Figure 1B.


21. Figure 2, the quality of this figure should be improved. The font size is too small to see, and the layout of the B. napus chromosomes is not good as well.

22. Figure 3, there are some format problems in this figure. Please check it carefully.

23. Figure 4, the quality of this figure is bad. The information in this figure is hard to see for the readers. And the spatial arrangement of each figure is not good. Please improve the quality of this figure, carefully. Similar problem was observed in Figure 5 and 6.

24. Figure 7, please state the meanings of “R” and “S” in the names of the samples. Moreover, “heat map” should be heatmap”.

25. The writing needs to be thoroughly checked and substantially improved. There are many typos and some inaccuracies that should be corrected. I strongly recommend language editing for this paper.

Experimental design

no comment

Validity of the findings

no comment

Additional comments

no comment

---

## Round 0.2 · Minor Revisions

The paper is significantly improved. However, the following points need to be addressed:

1. The title of the paper needs to be simplified. I suggest something like "Genome-wide evolution and expression analysis of MYB-CC gene family in three Brassica spp."

2. Figures 2, 3, and 4 are of low resolution and it is hard to see the information in them. Please provide high-resolution images

3. The English language needs to be improved

---

## Round 0.3 · Minor Revisions

Please provide a certificate after English language editing.

---

## Round 0.4 · Minor Revisions

Dear Corresponding Author,
One of the Section Editor has raised the following concerns. Please address them concerns before the final decision is made:

"In general the manuscript is well written and presented. A few minor modifications are suggested below in the Discussion thread.

At line 120 there is mention of the RNA-seq data; however, this is simply the raw data. In usual cases there is an assembly done with this data; however, if the data is used as produced this should be clarified within the manuscript. If assembly data is available, it should be placed in a third-party resource such as NCBI GenBank. Assembled transcriptomes can be deposited as a transcriptome shotgun assembly (GenBank TSA resource). Please see: https://www.ncbi.nlm.nih.gov/nuccore/ foe example.

There is also mention of reference genomes via their URLs around line 140; however, in many cases reference genomes are often updated so it is important to highlight the version numbers associated with the genomes referenced. In the section keyed on expression patterns via the transcriptome it is mentioned that data were collected, but the type of P influence associated within the studies is not discussed and it would be unclear how the authors can conclude the roles of the MYB sequences. It would be desirable if some added annotation can help extend the classification of the annotations via gene ontologies; the set is small so some mention or added fidelity to the data would be helpful in adding context.

The manuscript appears close to resolution; however, points of added clarity are still in need. Additional revision requested.

LINE NO: / BEFORE / AFTER / [COMMENTS]
LINE 39: / transfer, protein / transfer, and protein / [.]
LINE 59: / with an MYB / with a MYB / [.]
LINE 87: / Brassica rape / Brassica rapa / [.]
LINE 110: / petri dish / Petri dish / [.]
LINE 120: / . / . / [was there no assembly of RNAs? Just MYB matches?]
LINE 127: / by blasting in a local / using BLAST on a / [.]
LINE 139: / . / . / [Though sequences were derived from known reference websites, the version number of the reference should be known in case of updates.]
LINE 161: / . / . / [Was the transcriptomic data related to P influence in any of the data sets?]
LINE 196: / onto unanchored scaffolds. / . / [This suggests there is some assembly data that needs to be referenced.]
LINE 214: / B. rape / B. rapa / [.]"

---

## Round 0.5 · accepted · Accept

The paper is much improved and is acceptable in its current form.